# Motion Analysis of Football Kick Based on an IMU Sensor

**DOI:** 10.3390/s22166244

**Published:** 2022-08-19

**Authors:** Chun Yu, Ting-Yuan Huang, Hsi-Pin Ma

**Affiliations:** 1Interdisciplinary Program of Engineering, National Tsing Hua University, Hsinchu 300044, Taiwan; 2Department of Electrical Engineering, National Tsing Hua University, Hsinchu 300044, Taiwan; 3Center for Sport Science and Technology, National Tsing Hua University, Hsinchu 300044, Taiwan

**Keywords:** sports technology, football, motion analysis, IMU, trajectory reconstruction

## Abstract

A greater variety of technologies are being applied in sports and health with the advancement of technology, but most optoelectronic systems have strict environmental restrictions and are usually costly. To visualize and perform quantitative analysis on the football kick, we introduce a 3D motion analysis system based on a six-axis inertial measurement unit (IMU) to reconstruct the motion trajectory, in the meantime analyzing the velocity and the highest point of the foot during the backswing. We build a signal processing system in MATLAB and standardize the experimental process, allowing users to reconstruct the foot trajectory and obtain information about the motion within a short time. This paper presents a system that directly analyzes the instep kicking motion rather than recognizing different motions or obtaining biomechanical parameters. For the instep kicking motion of path length around 3.63 m, the root mean square error (RMSE) is about 0.07 m. The RMSE of the foot velocity is 0.034 m/s, which is around 0.45% of the maximum velocity. For the maximum velocity of the foot and the highest point of the backswing, the error is approximately 4% and 2.8%, respectively. With less complex hardware, our experimental results achieve excellent velocity accuracy.

## 1. Introduction

For any sports, repeated practice is required to improve performance and techniques. In addition to the amount of training, it is more important to use the correct method to enhance the quality of training. Practicing with improper methods is not only ineffective but also more likely to cause sports injuries. While performing a shot, players maximize speed and power, trying to make the shot more effective. However, for amateurs, exerting excessive force can easily lead to stiffness of the kicking leg. This results in insufficient knee bending which leads to momentum reduction during the foot swing before contacting with the ball. This problem is difficult to realize by athletes themselves. One way to analyze the motion is by applying multiple high-speed cameras combined with image analysis software to reconstruct the human body model and the state of motion. However, such equipment is relatively expensive and has environmental restrictions since image-related equipment needs to be set in a specific space or venue. On the other hand, IMU sensors have features such as light weight, low power, low cost and small size. An IMU can consist of a three-axis accelerometer, a three-axis gyroscope and a three-axis magnetometer. With proper filtering and data fusion, the information can be used for attitude and position estimation. Applications of IMU include military, automobile and sports.

### 1.1. Related Work

#### 1.1.1. IMU in Sports

Wearable sensors with IMUs have been utilized in pedestrian dead-reckoning systems by detecting the stationary stance phase and applying zero-velocity updates (ZUPTs) for position tracking [1]. Inertial sensors were placed on the side of the shoe in [2] to obtain information about foot clearance and mean step velocity, which helps assess foot kinematics in steady-state running. Another study [3] developed a system for field-based performance analysis based on IMUs which are attached to both ankles. The system detects stance duration, providing users with real-time feedback. In [4], the study used eight IMU sensors with velocity-based localization to capture the human spatial behavior and velocity during motions such as walking, jumping and running. The system was reduced to three IMU sensors and utilized the velocity-based localization with acceleration fine tuning [5].

To help prevent shoulder injuries, ref. [6] presented a classification approach by tracking and discriminating shoulder motions using an IMU. The wearable motion capture platform proposed in [7] provides physical quantities during the high-speed motion of baseball pitchers. With an array of inertial and magnetic sensors, the method allows for the analysis of various biomechanical parameters. A wearable device was developed by incorporating IMU sensors with flow sensors. The device in [8] measures human limbs velocity, acceleration and attitude angles. Experiments include boxing motion capture with the device on the forearm and kicking motion capture with the device on the shank. Ref. [9] presented a wearable sensing system consisting of multiple IMU sensors for basketball activity recognition. The system is able to identify walking, jogging, running, sprinting and shooting. Another basketball-related study built a wrist-worn sensor consisting of an IMU, five environmental sensors, a processor and a microcontroller. The activity recognition part was conducted by machine learning [10]. The algorithm proposed in [11] detects four key temporal events and three temporal phases in skateboarding. It can provide quantitative assessment for injury prevention.

#### 1.1.2. Football-Related Motion Analysis

Lower extremity and pelvis kinematics such as linear velocities and angular velocities were measured by an off-the-shelf product of 17 inertial sensors during kicking. The measurements were then compared with those obtained from an optoelectronic motion analysis system [12]. The hip joint motion of football players during practice was recorded directly on a sports field by a three IMU system [13]. The motion was characterized by hip acceleration and orientation. To quantify movement intensity and improve training load estimation, the system in [14] obtained knee and hip joint kinematics for football-specific movements performed at different intensities. A pressure-sensitive material was placed on the kicking foot in [15]. The device measured the force and center of pressure during the impact phase for players to further improve their technique. Biomechanical differences were observed during kicking with the preferred and the non-preferred leg [16]. Both kinetics and kinematics were derived from the filmed movements. By the full-body modeling and three-dimensional motion capture system, quantitative evaluations of kick quality were provided [17]. Using a single IMU and the acceleration data, the system in [18] distinguished between running and dribbling, passing and shooting. The study also compared three sensor locations (inside ankle, lower back and upper back) for better accuracy. Detection and segmentation of a soccer kick were performed by a system of wearable sensors and video cameras for sports motion analysis [19].

From the above paragraphs, most IMU-related motion analysis research focuses on activity classification or motion recognition during training or in a match. With the environmental limitations possessed by camera-based optoelectronic systems, the size and weight of IMU has a clear advantage. It is a popular choice when performing motion analysis. Although some research studies look at the motion itself, most of them dive into the information related to training load or biomechanical parameters on a specific joint or body part. In particular, no previous research reconstructed and analyzed the instep kicking motion with a single IMU. This paper aims to present a motion analysis system with increased accessibility, providing football players of all levels with instant feedback and an auxiliary training method to improve the instep kicking technique.

The field application of this study is expected to help players or general football lovers to adjust their movement posture before actual kicking. Preliminarily changing the posture in the empty kick stage will make the players develop good kicking habits more effectively, resulting in a better performance when actually kicking the ball. Therefore, this paper is mainly focusing on dealing with the trajectory of the foot during the kicking motion. The sensors are calibrated and the threshold setting is tailored for the kicking motion to avoid some tiny impact.

To validate the reliability of the system, we utilize the high-speed cameras to obtain the golden pattern for the trajectory. According to the systematic review study [20], one of the most commonly used measures of agreement is the Bland–Altman plot. It is a scatter plot which shows the relationship between two methods. The metric is used in this study to evaluate the accuracy of the trajectory reconstructed from the IMU data.

The system architecture is shown in Figure 1. We collect acceleration and angular velocity data during movement through the accelerometer and gyroscope in the six-axis inertial measurement unit (IMU). After the steps of deviation calibration, attitude estimation with quaternions, the transformation of coordinates, and gravity compensation, we analyze the maximum velocity and highest point of the foot before contacting with the ball while reconstructing the 2D and 3D trajectory of the kicking motion.

This paper aims to present a 3D motion analysis system which allows users to observe the kicking motion and acquire significant motion information, with only a single IMU sensor attached to the kicking foot, avoiding complex accessories which might affect training and eluding the hassle of setting up optoelectronic devices. The main contributions of this study are: (1) the synthesis of a simple motion trajectory reconstruction system for the data collected by a single six-axis IMU during an instep kicking motion, which employs the quaternion representation of orientation to describe the attitude change; (2) the customized adjustments to various parameters for the football kicking action during the signal processing, and the elimination of various possible noises to ensure that the accumulation of integral errors is minimized; (3) the extraction of specific motion data from the reconstructed trajectory to provide motion parameters that affect the quality of the kick during the process from backswing to kicking.

## 2. Methodology

The proposed sensing system includes data collection and several data processing procedures. More detailed steps will be given later in this chapter.

### 2.1. Data Collection and Deviation Calibration

The sensor selected in this research is ICM-20649 [21]. It is a wide-range six-axis motion tracking device which contains a three-axis accelerometer and a three-axis gyroscope, each with a 16-bit ADC, and the sampling frequency is set to 100 Hz. In the previous measurement, we found that the upper limit of the kicking motion is about 12 g, so we set the full signal range to ±30 g and ±4000 °/s for the application in this research. The precision measured from this range is acceptable because there are subsequent mechanisms for threshold and stationary judgement to distinguish the state of motion.

This experiment uses Bluetooth to transmit real-time data. After pairing the sensor with the Bluetooth receiver, the acceleration data and angular velocity data will be transferred to the computer and stored as text files. After the data are converted to decimal, it is necessary to perform the two’s complement to obtain the negative number.

A modified sphere model is applied in the calibration process for sensor deviation. First, we assume the calibration equation to be G = L(g + b), G is the acceleration before calibration, g is the real acceleration, L is the linear proportional deviation of the sensor itself and b is the deviation of the center value of the sensor. In an ideal static state, the sum of the squares of the three-axis acceleration should be equal to one, so the gravitational acceleration values at various angles will form a sphere with a radius of one. When calculating, one must first assume that the linear proportional deviation is one, and one must use the least square method to obtain the center of the three axes. The same method can be used to find the linear proportional deviation, but the actual test found that the three-axis acceleration square sum will be less than one when the sensor is stationary. Therefore, normalization is performed in the end to complete the accelerometer calibration.

### 2.2. Attitude Estimation with Quaternion

In the common state of motion, rotation is bound to participate, and the acceleration received by the three axes of the sensor is actually the acceleration of the sensor’s coordinates, not the acceleration of the earth coordinates. Data can only be applied and analyzed through attitude processing. The six-axis sensor chosen for this research only includes an accelerometer and a gyroscope. Without a magnetometer, we can only obtain the sensor attitude by obtaining the respective angle changes of the sensor and comparing them with that of the initial coordinates.

Quaternion representation of rotation is derived from the characteristics of inner and outer products between vectors. It can be considered to be the extension of two-dimensional real and imaginary numbers to four-dimensional to show the rotation in three-dimensional space. Similar to complex numbers, quaternions are composed of real numbers and three elements *i*, *j* and *k*. Each quaternion q can be represented by a linear combination of them, generally expressed as q=a+bi+cj+dk, and they follow the following relationship:(1)$$i^{2}=j^{2}=k^{2}=ijk=-1$$

The attitude quaternion (*q*) is a column vector of four parameters to describe a rotation along a specific axis, which can be written as:(2)$$q=\left[\begin{array}{c} q_{0}\\ \begin{array}{c} q_{x}\\ q_{y} \end{array}\\ q_{z} \end{array}\right]\;≜\;\left[\begin{array}{c} \cos\left(\frac{\theta }{2}\right)\\ \begin{array}{c} E_{x}\sin\left(\frac{\theta }{2}\right)\\ E_{y}\sin\left(\frac{\theta }{2}\right) \end{array}\\ E_{z}\sin\left(\frac{\theta }{2}\right) \end{array}\right]$$

However, in a general movement, it is difficult to know the rotation axis of each sampling point, and the angle information is of the sensor axes instead of the axis with which the sensor rotates along. Since the angle information is obtained through the gyroscope, we decide to directly update the quaternion by using the angular velocity data. The vector $S_{\omega }$ which contains the angular velocities is defined as:(3)$$S_{\omega }=\left[0\;\;\;\;\omega_{x}\;\;\;\;\omega_{y}\;\;\;\;\omega_{z}\right]\;\;\;$$

Then, we consider the quaternion derivative that describes the rate of change in orientation:(4)$$\frac{dQ_{k}}{dt}=\frac{1}{2}\cdot {\hat{Q}}_{k-1}⨂S_{\omega }$$

The first parameter, $\frac{dQ_{k}}{dt}$, is the derivative at time step *k* expressed in quaternion, ${\hat{Q}}_{k-1}$ is the estimated orientation at time step k, and ⨂ is the quaternion product operator. By integrating the quaternion derivative, it would be possible to estimate the orientation over time:(5)$${\hat{Q}}_{k}={\hat{Q}}_{k-1}+\frac{dQ_{k}}{dt}\;\cdot \;\Delta t\;$$

Finally, we can use the following equation to complete the quaternion update:(6)$$Q_{k}+=0.5\times Q_{k-1}\times angVel\times dt$$

In addition, after each update of the quaternion, the quaternion must be normalized to obtain the true quaternion, so as to avoid the phenomenon of scaling while the vector is rotating. When a new quaternion is obtained, the acceleration data of the sensor can be converted into the acceleration data of the initial coordinates through the following formula:(7)$$accl_{transformed}=Q\times accl\times Q_{conj}$$
where $accl_{transformed}$ is the acceleration data in initial coordinates, accl is the acceleration data before attitude processing, Q and $Q_{conj}$ represent the quaternion and the conjugate quaternion, respectively.

### 2.3. Gravity Compensation

This subsection will introduce the method of compensating the gravity components and the transformation of coordinates. Since the sensor data during the entire motion have been converted into the initial sensor coordinates, we can subtract the average acceleration of the first 500 sampling points obtained in the static state offset_accl from the raw acceleration data. Through this process, we can obtain the movement data of the sensor without the influence of gravity.

After gravity compensation, the misalignment between the initial coordinates and the earth coordinate still needs to be dealt with. If this problem remains unsolved, the 2D and 3D motion trajectory will be tilted. Different from the previous processing of attitude changes, since the initial coordinates are those at rest and cannot be processed with angular velocity information, we implement the rotation matrix of the initial coordinates to the earth coordinates to calculate the inclination of the gravity component.

First, we divide the rotation into three parts: roll, pitch and yaw. The tilt of a three-dimensional space can be achieved with two axial rotations.
(8)$$\mathrm{roll}=\arctan\left(\frac{{\mathrm{offset}}_{y}}{{\mathrm{offset}}_{z}}\right),\;\;\;\;\mathrm{pitch}=-\arctan\left(\frac{{\mathrm{offset}}_{x}}{{\mathrm{offset}}_{z}}\right),\;\;\;\;\mathrm{yaw}=0$$

After obtaining the rotation angles around each axis, we find the rotation matrix, and combine the three with matrix multiplication to obtain the complete rotation matrix, written as the following matrices:(9)$$R_{x}=\left[\begin{array}{ccc} 1 & 0 & 0\\ 0 & \cos\left(\mathrm{roll}\right) & -\sin\left(\mathrm{roll}\right)\\ 0 & \sin\left(\mathrm{roll}\right) & \cos\left(\mathrm{roll}\right) \end{array}\right]\;\;\;\;\;\;\;\;R_{y}=\left[\begin{array}{ccc} \cos\left(\mathrm{pitch}\right) & 0 & \sin\left(\mathrm{pitch}\right)\\ 0 & 1 & 0\\ -\sin\left(\mathrm{pitch}\right) & 0 & \cos\left(\mathrm{pitch}\right) \end{array}\right]\;\;R_{z}=\left[\begin{array}{ccc} \cos\left(\mathrm{yaw}\right) & -\sin\left(\mathrm{yaw}\right) & 0\\ \sin\left(\mathrm{yaw}\right) & \cos\left(\mathrm{yaw}\right) & 0\\ 0 & 0 & 1 \end{array}\right]\;\;\;\;\;T_{rotate}=R_{z}\times R_{y}\times R_{x}\;\;$$

Lastly, we multiply it by the three-axis acceleration after compensating the gravity to complete the transformation of the coordinates, written as:(10)$$accl_{corrected}=T_{rotate}\times accl_{corrected}$$

### 2.4. Quadratic Integration and Threshold Setting

After completing the transformation of the coordinates and the gravity compensation, we proceed to the trajectory construction part. The velocity can be obtained by integrating the acceleration once, and the displacement can be obtained after the second integration. The displacement between every two sampling points can be used to reconstruct the trajectory of the sensor movement.

In this research, we slightly modified the integration method by averaging the acceleration value between two sampling points to calculate the acceleration value belonging to the time interval. The formula can be written as:(11)$$v_{i}=v_{i-1}+\frac{a_{i}+a_{i-1}\;\;}{2}\;\Delta t$$

The result calculated by this integration method is more accurate than that calculated by the original formula $v_{i}=v_{i-1}+a_{i}\Delta t$. The velocity change, which is the area calculated by this method, is shown by the area $a^{\prime }\Delta t$ in Figure 2, and $a^{\prime }$ is the average acceleration of $a_{1}$ and the acceleration from the previous sampling point. It can be found that the purple area on the left can be roughly compensated to the original missing area, so the integral error will be smaller than the original formula. We perform the integration separately on the three-axis data collected by the sensor to obtain the velocity of each axis, and then we use a similar integration technique to obtain the displacement.

Threshold setting is a crucial aspect when integrating. During the experiment, the sensor will inevitably be affected by some external factors, such as vibration, wind and incomplete compensation of gravity components. The slight fluctuation of acceleration has a considerable influence on the error of the integration. Therefore, after repeating several experiments, we found that the acceleration of the target motion is mostly above $3.92\;m/s^{2}$. We set $0.392\;m/s^{2}$ as the acceleration threshold to filter the acceleration value of the target movement before integration.

In addition, there will be a physical blind spot in the actual acceleration integration. When the sensor is stationary after a motion, the acceleration integration area during acceleration and deceleration cannot completely offset each other. Even if the sensor is at rest and the acceleration has become exactly zero, the velocity remains at the same value of the previous sampling point. In this case, when the velocity is integrated to obtain the displacement, the sensor will seem to continue its motion at a constant velocity instead of being in a static state. Therefore, a new judgment condition is added here. When the acceleration of fifteen consecutive sampling points is zero, it is determined to be a static state, and the velocity is returned to zero. A reasonable velocity threshold is also obtained through multiple experiments, and is set to 0.196 m/s to ensure that the above-mentioned accumulation of errors will not occur.

## 3. Results

### 3.1. Experimental Setup

Two high-speed cameras are used to capture the image from the front and side view to provide golden patterns for the experiment; we use tripods to secure the camera to avoid shaking, and place multiple scale bars within the capture range as a reference for depth correction. After setting up the cameras, we tie the sensor (ICM-20649) on the top of the athlete’s foot with a rubber band, and perform an instep kicking motion without hitting a ball. The data received from the IMU will be collected and imported to MATLAB for data processing, then we draw trajectory diagrams and analyze different motion data.

The theoretical value of the experiment is provided by the video of the cameras. We import the video into Tracker for mapping and export the 2D data of each angle of view, align the peaks through the front view and the side view, and then perform the depth correction separately. The 3D data can be combined and the data can also be imported into MATLAB as the theoretical values. The results will then be used to calculate the error of each analysis.

### 3.2. Experimental Results

#### Motion Trajectory Analysis

After completing the data processing introduced in the previous chapter, the 3D position information of each sampling point of the IMU will be obtained, and the 3D trajectory diagram will be drawn with MATLAB. The average path length in several repeating experiments and the root mean square error (RMSE) with the theoretical value of the entire path will be calculated to verify the accuracy of the system. The two trajectories are aligned from the beginning of the motion, and then we utilize the relative sampling rate according to the different sample rates of the IMU and the frame rate of the camera. We calculate the distance between the corresponding sample points and calculate the RMSE of the position and the velocity in the direction of the kick. Figure 3 is a 3D motion trajectory diagram, the blue solid line in the figure is the theoretical trajectory obtained by Tracker, and the line composed of the red dots is the trajectory obtained after IMU data are processed.

### 3.3. Foot Velocity Analysis

On the football field, whether it is passing or shooting, the velocity of the ball is a crucial factor. We hope to observe the maximum velocity of the athlete’s foot swing and where the maximum value occurs so that we can help athletes transmit the most kinetic energy to the ball. With the golden pattern obtained by Tracker, we can compare the velocity of the sensor with the velocity from the video. Figure 4 is a 2D motion trajectory diagram, the blue cross is the position where the maximum velocity appears in the theoretical trajectory, and the red circle is the position where the maximum velocity appears in the IMU motion trajectory.

### 3.4. Backswing Height Analysis

When shooting or hitting a long ball, if the knee of the kicking foot is not bent enough to increase the height of the foot, the power of the ball will be significantly affected. Therefore, we would like to observe the height of the highest point of the foot during the pull-back motion on the reconstructed trajectory. With the golden patterns obtained by Tracker, we can discuss the accuracy of the system by comparing the highest points during the backswing. We can also use the 3D trajectory graph to obtain the position of the highest point for visualization. Figure 5 is the 3D motion trajectory diagram and the highest point of the backswing.

Table 1 shows the quantified results generated from IMU data and also the results from high-speed cameras. From the results below, we can observe that in the motion with an average path of about 3.6 m, the entire trajectory obtained by IMU’s data processing with the theoretical trajectory only has an absolute error of about 0.07 m. It is considered a very accurate result when constructing a motion trajectory, thus it proves that our signal processing system has a certain degree of credibility. As for the instantaneous velocity of the foot and the backswing height, the error is approximately 4% and 2.8%, respectively.

Figure 6 shows the validation of position (three axes) during the motion by comparing the IMU algorithm results with high-speed camera results. From the Bland–Altman plot, it can be seen that only 4.17% (10 out of 240) of the points are outside the 95% limits of agreement, the extent of the difference is clinically acceptable, so the two methods can be considered to be in good agreement, inferring that this IMU algorithm can be clinically substituted for high-speed camera.

## 4. Discussion

While camera-based optoelectronic systems can provide high accuracy for motion capturing, it has environmental restrictions and has limitations in capture rate. When calculating derivatives greater than or equal to second order using the measurement data, it has a high level of noise, often resulting in limited or no physical meaning unless the raw data are filtered to 10–20 Hz [22]. When these optoelectronic systems are applied to targets moving in high speed, although the position will be accurate, the velocity and acceleration might not be of adequate accuracy. At the same time, the device settings of these image analysis systems are cumbersome and can only be used in a specific environment. The IMU sensor is undoubtedly a perfect substitute in this case. It provides the information of inertial data such as acceleration and angular velocity directly. The sensor can be easily mounted on the person without interfering with their performance, and since the sensor is light, players can easily adapt to the existence of the new device. 

Focusing on the football kicking motion, we constructed a motion analysis system based on an IMU sensor, trying to analyze the physical quantities related to improving the football kicking performance. To preliminarily evaluate and assess a kicking motion, the foot velocity and backswing phase are both key factors related to the quality of the kick. In [23], the results showed that the foot velocity at the initial instant at the initial impact phase affects the ball velocity more than any other factors. The quality of foot–ball contact is crucial to the spin and speed of the ball. Higher foot velocity is related to more powerful kicks [24]. 

For the reconstructed trajectory, our system has achieved results with high accuracy and low RMSE in both position and velocity. Since the types of target motions are different, it sometimes cannot fully explain whether a method outplays another simply by comparing the RMSE without considering the length of the motion and the dimensions evaluated. For the gait analysis algorithm that Zhou et al. performed in [25] on the action of striding forward, they achieved an RMSE of about 0.05 m in a stride of about 1.5 m. As for the acceleration-based simultaneous localization and capture method (A-SLAC) proposed in [5], the RMSE in the main walking direction is 0.038 m for a length of 3.6 m for each trial. The RMSE is 0.032 m for the vertical direction and 0.057 m in the sideways direction, which is about 2% of the trial length. While they focus on performing the error calculation on the direction of the stride, we conduct the error calculation of the 3D motion. For an instep kicking motion with the average path length of around 3.63 m, our system achieved the position RMSE of 0.07 m.

For velocity, we extracted the maximum instantaneous velocity from the kick; the results showed a 4% error compared to the image captured by the high-speed cameras. Moreover, the RMSE of the foot velocity is about 0.034 m/s, which is around 0.45% of the maximum velocity (7.47 m/s). For the velocity in the main walking direction in [5], the RMSE is 0.051 m/s, which is around 3% of the maximum velocity (1.5 m/s). The results indicate our system performs better in the accuracy of velocity. Table 2 shows the accuracy evaluation results obtained for different types of motion using different IMU-based systems.

With the steady evolvement of wearable IMUs, inertial components are now commonly integrated onto a single die, allowing users to receive various motion-related data. The development of high-resolution and wide-range devices would be ideal for measuring motion poses in high-intensity motion. Moreover, the stretchable electronics would enable devices with multiple sensors to be embedded into forms that are more suitable for mounting on the body [26,27,28]. Multiple inertial sensor nodes would even provide better motion tracking; since there are more data, we can use the gradient descent method to fuse data and obtain a more accurate trajectory [29]. Moreover, by fusing the position and orientation data from the optoelectronic systems with the inertial data obtained from the IMU, we might be able to obtain the best set of kinematics data. By applying sensor fusion techniques based on a multiple-model linear Kalman filter for deflection estimation, the data can be fused with low processing cost, compatible with real-time embedded applications [30].

## 5. Conclusions

For the motion analysis, we develop a data processing procedure to fuse data from the accelerometer and gyroscope of the IMU. According to the experiment results, for the instep kicking motion of trajectory length around 3.63 m, the root mean square error of the position and the velocity compared with the golden patterns obtained from the high-speed cameras and image analysis software is about 0.07 m and 0.034 m/s, respectively. For the maximum velocity of the foot, the error is approximately 4%. This metric is related to the contact point with the ball and the timing of acceleration. The error for the highest point of the foot before hitting the ball is 2.8%.

This system can be applied to players of all ages and levels, whether it is to observe movement changes by trajectory, or simply to measure the height or velocity of the feet. The motion information provided in the quantified form allows players or coaches to have a more specific and clear method to analyze the action. The experiment in this research does not require a large amount of equipment, nor does it need to be carried out in a specific place or room, hence the convenience of practical application is greatly improved.

## Figures and Tables

**Figure 1 sensors-22-06244-f001:**
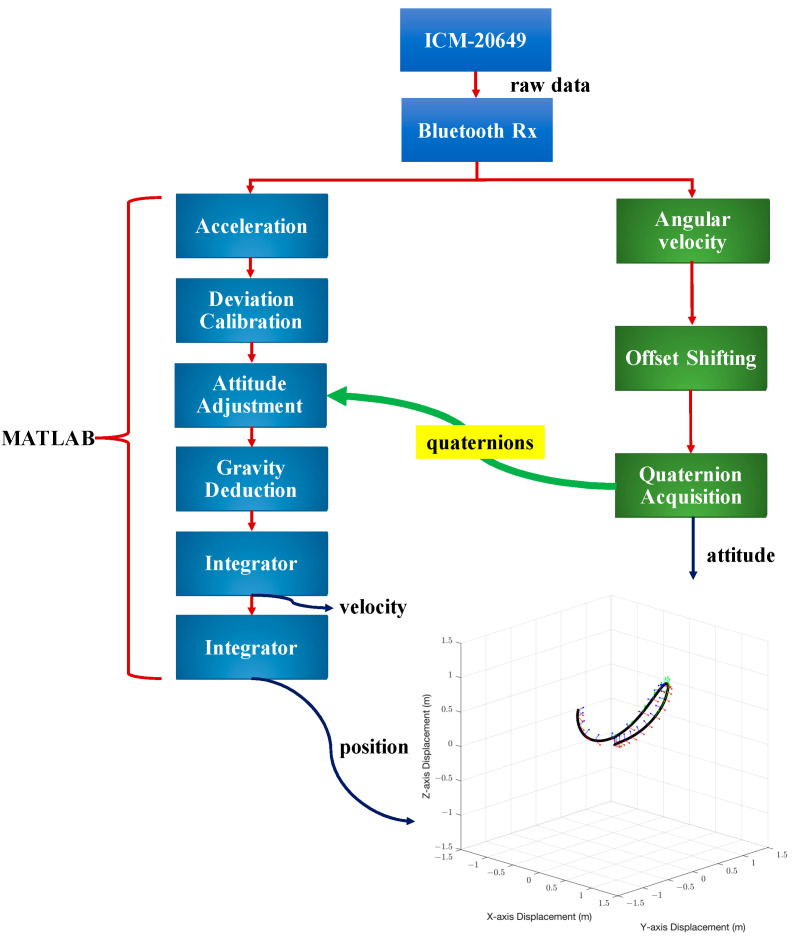
An illustration of the proposed system. The three colored small axes on the trajectory represent the coordinates of the sensor.

**Figure 2 sensors-22-06244-f002:**
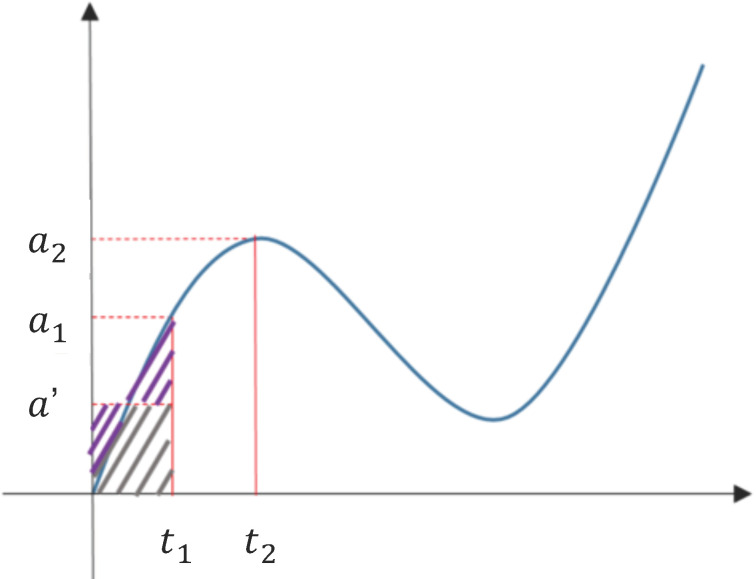
Illustration of integration error cancellation. The average acceleration of two adjacent sampling points is taken for calculation.

**Figure 3 sensors-22-06244-f003:**
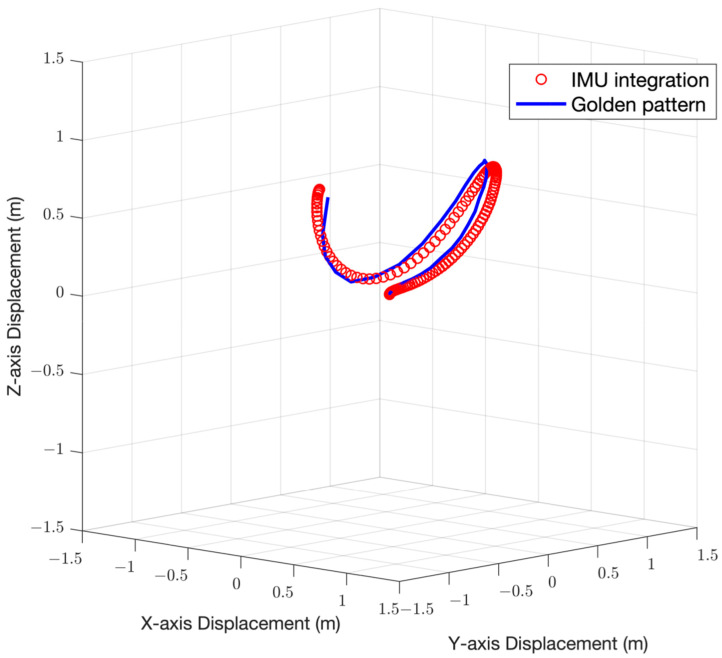
Three-dimensional motion trajectory diagram. For an instep kicking motion of path length around 3.63 m, the position RMSE and the velocity RMSE of the two trajectories are 0.07 m and 0.034 m/s, respectively.

**Figure 4 sensors-22-06244-f004:**
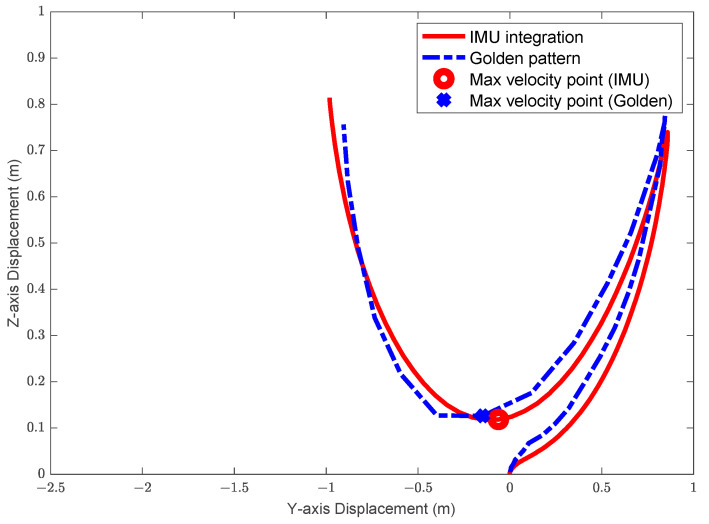
Two-dimensional trajectory diagram with maximum velocity position. The maximum velocity occurs when the foot reaches the bottom of the motion trajectory. An average value of the maximum instantaneous velocity in repeated experiments is around 7.4 m/s, and an error of 4% is achieved.

**Figure 5 sensors-22-06244-f005:**
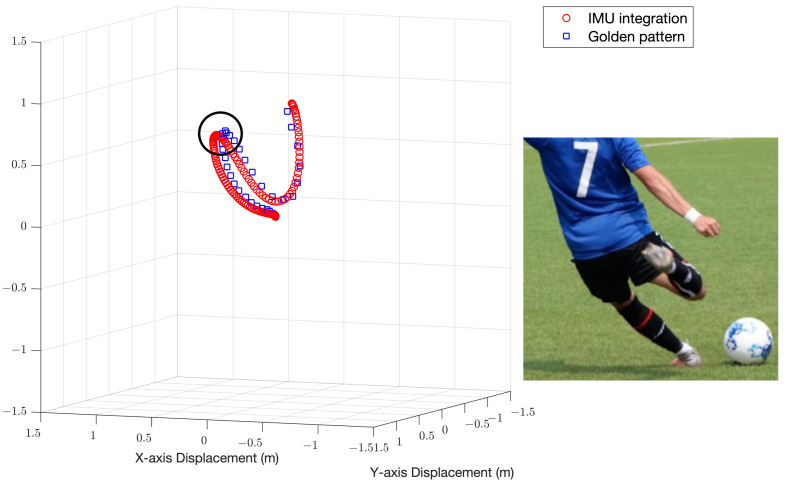
Three-dimensional trajectory diagram with backswing height illustrated. An error of 2.8% is achieved for an average backswing height of 0.756 m. The image on the right shows the highest point during the backswing.

**Figure 6 sensors-22-06244-f006:**
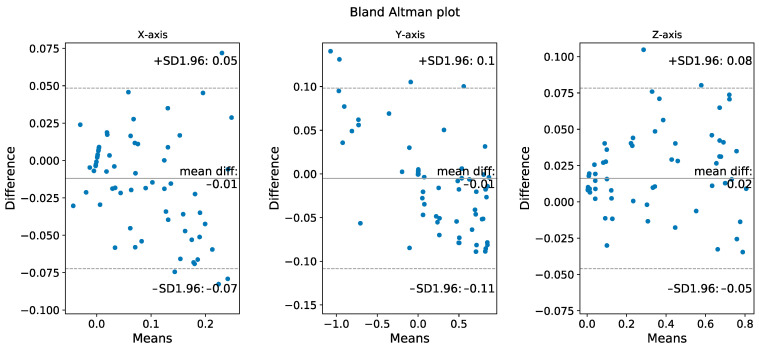
Validation of position during the motion by comparing the IMU algorithm results with high-speed camera results. The Bland–Altman plots for the three axes show that the data obtained by these two methods have high similarity.

**Table 1 sensors-22-06244-t001:** Comparison of reconstructed trajectory, instantaneous velocity and backswing height generated by IMU data with high-speed cameras’ results.

**Instep Kicking Test of Sample Size 10**	**Reconstructed Trajectory**			**Foot Velocity Analysis** **(Instantaneous Velocity)**			**Backswing Height Analysis**		
	**Average Length (m)**	**Position** **RMSE (m)**	**Velocity** **RMSE (m/s)**	**IMU** **(m/s)**	**Image** **Analysis (m/s)**	**Error**	**IMU** **(m)**	**Image** **Analysis (m)**	**Error**
	3.63	0.07	0.034	7.468	7.409	4.0%	0.741	0.756	2.8%

**Table 2 sensors-22-06244-t002:** Accuracy evaluation results obtained for different types of motion using different IMU-based systems. For the position RMSE in the gait-related system and A-SLAC system, the error is calculated according to the direction of movement, while our system performs it with the 3D trajectory.

	Motion Type	Motion Length	Position RMSE	Maximum Velocity	Velocity RMSE	Velocity RMSE %	IMU Used
Gait-related	stride	1.5 m	0.05 m	N/A	N/A	N/A	2
A-SLAC	walking	3.6 m	0.038 m	1.5 m/s	0.051 m/s	3%	3
Our system	instep kicking	3.63 m	0.07 m	7.47 m/s	0.034 m/s	0.45%	1

## Data Availability

The data presented in this study are available on request from the corresponding author.

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
