# Peer review of "Motion Analysis of Football Kick Based on an IMU Sensor"

_sensors, 2022, doi:10.3390/s22166244_

Round 1
Reviewer 1 Report
In this study, an IMU was used to analyze the kick motion of soccer that does not impact the actual soccer ball.
I have a few questions about this study.
1. Is there any reason not to analyze the actual kicking motion? The data of the inertial sensor is greatly affected by the mass and the amount of impact. If these are not calibrated, it may be more advantageous to analyze the motion using a conventional vision sensor.
2. Is the IMU data analysis method different from the existing IMU analysis method in this study? Newer and more calibrated methods require results that compare with traditional IMU analysis methods.
3. The camera-based motion analysis method also has errors. Comparing this method with the IMU method, the actual error may be more significant if there is an error. Therefore, it is necessary to verify the accuracy through the actual measurable motion (using a jig or gimbal) before analyzing the actual motion.
I think questions 1 and 2 are critical. In particular, if Errors are not corrected at the time of impact, the IMU cannot be used to analyze soccer motion.
Please reply to comments and questions.
Author Response
Point 1: Is there any reason not to analyze the actual kicking motion? The data of the inertial sensor is greatly affected by the mass and the amount of impact. If these are not calibrated, it may be more advantageous to analyze the motion using a conventional vision sensor.
Response 1: This is an important question, thank you for asking it, letting us know that we were not clear enough in our original explanation. The field application of this study is expected to help players or general football lovers to adjust their movement posture before actual kicking. Preliminarily changing the posture in the empty kick stage will make the players develop good kicking habits more effectively, resulting in a better performance when actually kicking the ball. Therefore, this paper mainly focuses on dealing with the trajectory of the foot during the kicking motion. The sensors are calibrated and the threshold setting is tailored for the kicking motion to avoid some tiny impact. We have added the explanation into the manuscript in line 100-106 in the introduction section.
Point 2: Is the IMU data analysis method different from the existing IMU analysis method in this study? Newer and more calibrated methods require results that compare with traditional IMU analysis methods.
Response 2: In most of the existing IMU related motion analysis works, they mainly focus on motion recognition or obtaining information related to training load or biomechanical parameters on a specific joint or body part. Few studies focus directly on the 3D trajectory and no previous research looks at the reconstruction of the instep kicking motion. The motion analysis method used in this study aims to provide a clear trajectory plot with position and velocity of each point of the process, the maximum instantaneous speed of the foot, and the backswing height be easily assessed, allowing users to compare it with themselves or professionals in a quantified way.
In the revised version of the manuscript, the RMSE of the velocity is also introduced, and we’ve achieved 0.034 m/s error, which is about 0.45% of the maximum instantaneous speed. We also use Bland-Altman plot to show the accuracy of our system. The results show that there is a strong relation between our method and the camera-based method.
Point 3: The camera-based motion analysis method also has errors. Comparing this method with the IMU method, the actual error may be more significant if there is an error. Therefore, it is necessary to verify the accuracy through the actual measurable motion (using a jig or gimbal) before analyzing the actual motion.
Response 3: In order to minimize the potential errors, the two high-speed cameras are used to capture image from both the front and side view to provide golden patterns for the experiment, we used tripods to secure the camera to avoid being shaked, and we placed multiple scale bars within the capture range as a reference for depth correction. Before actually using it as a golden standard, we have also done experiments such as linear motion and vertical drop test to verify the reliability of the method. We believe that this camera-based method can provide a golden pattern for the comparison of our system. To validate the reliability of the system, we utilize the high-speed cameras to obtain the golden pattern for the trajectory. According to the systematic review study [20], one of the most commonly used measurement of agreement is the Bland-Altman plot. It is a scatter plot which shows the relationship between two methods. The metric is used in this study to evaluate the accuracy of the trajectory reconstructed from the IMU data. The revised part in the manuscript can be found in line 107-111 (in section Introduction) and 331-340 (in section Results).

Reviewer 2 Report
This paper proposed a IMU-based motion analyzing system for football kick. The methods are simply described and the results are given. However, I do not think the paper can be published at the current state. The following is my comments.
1. About the motivation. The introduction part did not show the reasons to do this work and the importance of this work. It should be clearly shown in the Introduction.
2. About the novelty. I cannot get the novelty of this work. 1st, the Introduction did not show their highlights and then the core methods mentioned in the paper are all based on the common operations. The results are not greatly improved, when compared with other papers.
3. About the IMU device. Only the model is mentioned, no further information is provided. At least, a citation for the datasheet of this IMU is needed. Then, will the precision of IMU influence the measured results? How to get rid of this influence?
4. What is the theoretical trajectory in Line 218 and ideal trajectory in Fig.3?
5. Fig3 states ”For an instep kicking motion of path length around 3.63 m, 222
the RMSE of the two trajectories is about 7 cm. “ How to get these values (especially 7cm) ?Also, 7.4m/s in Fig.4?
6.A comprehensive comparison between the results and published ones is needed.
Author Response
Point 1: About the motivation. The introduction part did not show the reasons to do this work and the importance of this work. It should be clearly shown in the Introduction.
Response 1: This paper aims to present a 3D motion analysis system which allows users to observe the kicking motion and acquire significant motion information, with only a single IMU sensor attached to the kicking foot, avoiding complex accessories which might affect training and eluding the hassle of setting up optoelectronic devices. We have modified the manuscript to show the motivation and importance of the proposed work. Please see the revised part in the introduction.
Point 2: About the novelty. I cannot get the novelty of this work. 1st, the Introduction did not show their highlights and then the core methods mentioned in the paper are all based on the common operations. The results are not greatly improved, when compared with other papers.
Response 2: Many previous studies related to IMU are used for gait analysis or use multiple sensors to analyze the posture of athletes for identification, but no relevant experiments have been seen to focus on the reconstructed trajectory of the foot and analysis of it during kicking. The movement in the first half of the kicking action (backswing) actually has a significant impact on the kicking effect. We used some existing methods, adjusted them according to the backswing motion and integrated them together. We didn’t apply the existing AHRS method, but instead we directly used the angular velocity to update the quaternions and do the acceleration transformation, and surprisingly the results had no big difference from that of the complete AHRS method. Athletes can know the position and value of the maximum foot speed, and they can know whether the timing of exerting force is too early or too late through the graphical presentation. In addition, the influence of the degree of backswing on the kick is often overlooked. The intensity of a kick is largely affected by the swing of the backswing, and the stiffness of the calf is the reason behind it. By presenting the 3D trajectory and providing quantitative height information for the motion, the player can speculate whether the kick has a more severe stiffness problem than other kicks. We have modified the manuscript to add the contributions in the Introduction section, and provided some comprehensive analysis and comparison with related works in the Sec. 4 Discussion.
Point 3: About the IMU device. Only the model is mentioned, no further information is provided. At least, a citation for the datasheet of this IMU is needed. Then, will the precision of IMU influence the measured results? How to get rid of this influence?
Response 3: We have added the information about the IMU used in the experiment to the manuscript. The sensor selected in this research is ICM-20649. It is a wide-range 6-axis motion tracking device which contains a three-axis accelerometer and a three-axis gyroscope, each with a 16-bit ADC, and the sampling frequency is set to 100 Hz. In the previous measurement, we found that the upper limit of the kicking motion is about a dozen g, so we set the full signal range to ±30g and ±4000 °/s for the application in this research. The precision measured from this range is acceptable, because there are subsequent mechanisms for threshold and stationary judgement to distinguish the state of motion. Please see the revised part in Sec. 2.1.
Point 4: What is the theoretical trajectory in Line 218 and ideal trajectory in Fig.3?
Response 4: The theoretical trajectory in Line 218 and ideal trajectory in Fig.3 were meant to be the same thing which, infers the golden pattern, is the trajectory obtained by the camera-based motion analysis method. We made a mistake for not unifying the proper nouns, now they are all changed to the term “golden pattern”, the term appeared in the entire article and in the chart are now unified. Sorry again for causing the misunderstanding.
Point 5: Fig3 states ”For an instep kicking motion of path length around 3.63 m, 222 the RMSE of the two trajectories is about 7 cm. “ How to get these values (especially 7cm) ?Also, 7.4m/s in Fig.4?
Response 5: We utilize the relative sampling rate according to the different sample rate of the IMU and the frame rate of the camera, and from the aligned sample points, we calculate the distance of the corresponding points and do the position root mean square error calculation throughout the trajectory. In the revised version of the manuscript, we also introduced the velocity RMSE, which is calculated with similar method as the position RMSE. As for the 7.4 m/s in figure 4, that refers to the maximum instantaneous speed of the kicking. We take that out and compare because we want to know the value of the maximum foot speed when kicking the ball, and also know the timing it occurs.
Point 6: A comprehensive comparison between the results and published ones is needed.
Response 6: We have added the comprehensive analysis and comparison with related works. Please see the revised part in Sec. 4 Discussion.

Round 2
Reviewer 1 Report
I think this manuscript has been well revised.
It would be good if things like the actual kicking motion were carried out in a follow-up study.
Reviewer 2 Report
Thank you for your works, and the paper has been greatly improved after this round of revision. I think it can be accepted now.